# Machinability Study of Hardened 1045 Steel When Milling with Ceramic Cutting Inserts

**DOI:** 10.3390/ma12233974

**Published:** 2019-11-30

**Authors:** Mohamed Shnfir, Oluwole A. Olufayo, Walid Jomaa, Victor Songmene

**Affiliations:** 1Department of Mechanical Engineering, École de Technologie Supérieure (ÉTS), 1100 Notre-Dame St. West, Montréal, QC H3C 1K3, Canada; mohshnfir@gmail.com (M.S.); oluwole-ayodeji.olufayo.1@ens.etsmtl.ca (O.A.O.); walid.jomaa@cegepmontpetit.ca (W.J.); 2Cégep Édouard-Montpetit, Centre technologique en aérospatiale (CTA), 5555 Place de la Savane, Saint-Hubert, Québec, QC J3Y 8Y, Canada

**Keywords:** machinability, hard milling, AISI 1045 steel, ceramic tools, hardness, edge preparation

## Abstract

Intermittent machining using ceramic tools such as hard milling is a challenging task due to the severe mechanical shock that the inserts undergo during machining and the brittleness of ceramic inserts. This study investigates the machinability of hardened steel AISI 1045 during face milling using SiAlON and whisker (SiCW) based ceramic inserts. The main focus seeks to identify the effects of cutting parameters, milling configuration, edge preparation and work material hardness on machinability indicators such as resultant cutting force, power consumption and flank tool wear. The effects of these varying cutting conditions on performance characteristics were investigated using a Taguchi orthogonal array design L32 (2^1^ 4^4^) and evaluated using ANOVA. Results indicate lower resultant cutting forces were recorded with honed edge inserts of SiAlON ceramic grade. In addition, a decrease in resultant cutting forces was associated with reduced feed rates and increased hardness. The feed rate and cutting speed were also identified as the greatest influencing factors in the amount of cutting power. The main wear mechanisms responsible for flank wear on the ceramic inserts are micro-scale abrasion and micro-chipping. Increased flank wear was observed at low cutting speed and high feed rates, while micro-chipping mostly ensued from the cyclic loading of the radial tool edge form, which is more susceptible to impact fragmentation. Thus, the use of tools with chamfered tool-edge preparation greatly improved observed wear values. Additional confirmation tests were also conducted to validate the results of the tests.

## 1. Introduction

Due to the development of high-performance machine tools and advanced cutting tool materials, hard machining technology was accepted as a reliable technique in the manufacturing of structural components made of hardened materials (hardness exceeding 45 HRC). Many aspects of the hard machining processes such as parts’ surface integrity [1,2], machinability indexes [3,4] and ecological trends [5] have been extensively studied. It was demonstrated that these aspects were significantly sensitive to various technological parameters and their interactions [6]. In addition to the standard machine-tool parameters (cutting speed, feed rate and depth of cut), the cutting tool edge preparation and material, the workpiece material hardness and/or microstructure are amongst the critical factors that should be considered in the optimization of the hard machining processes, particularly, during milling operations (intermittent cutting). Cutting tool manufacturers have designed different cutting edge micro-geometries in order to withstand the high cutting pressure and reduce the catastrophic failure of the cutting tool during the machining of difficult-to-cut materials [7]. The edge preparations were principally applied to advanced cutting tools made of ceramic, cubic boron nitride (CBN) and polycrystalline cubic boron nitride (PCBN) tools. Compared to hard turning process, milling of hardened steel using advanced cutting tools was less attractive over the last two decades [1,2,3,4]. Elbestawi et al. [8] investigated the milling of AISI H13 steel (HRC 55) using PCBN ball-nose end mills with various edge preparations. Results showed that the worst performance was noticed for the honed edge in comparison to sharp and chamfered edge preparations. More recent studies by Wojciechowski S. have investigated the machinability of hardened steels when using ball-nose end mills [9,10]. They found that when milling with a flexible ball end nose tool, the obtained surface roughness is very sensitive to the selection of the cutting speed while in the case of milling with a rigid tool, the surface roughness depends less on the cutting speed used. The authors explained the first phenomena by the loss of rigidity, radial runout and chatter [10]. This effect of the chatter and run-out are very important when using ceramic inserts that are very sensitive to vibrations and fluctuations on cutting forces.

Wojciechowski S. [11] proposed a model for estimating the cutting forces when milling inclined parts with ball end nose tools and showed that the cutting forces coefficients depend on the tool inclination relative to the cutting direction and that small radial run out value (as small as 3 μm) can cause considerable cutting force variations during finish milling with ball tool. This result shows the importance of balancing the cutter prior to the milling process. Although the proposed model is more accurate, it could not be applied directly (without some modifications) to rigid milling with the tool having round inserts.

To lower machining cost by using cheaper cutting tools, researchers investigated hard machining using carbide tools with advanced coating and/or specific edge preparation. Li et al. [11] studied the effect of the edge hone radius in hard milling of AISI H13 steel (50 HRC) using cemented carbide inserts. The authors found that when edge radius increases, cutting forces, plastic deformation and compressive residual stress increase whereas the surface roughness decreases to a certain limiting value. Another study by Li et al. [12], investigating the hard milling of AISI H13 steel explored the effect of edge hone radius on the chip formation mechanisms. The results showed that when the edge hone radius increases, the chip segmentation intensity and frequency increases, leading to high cutting force fluctuations. Denkena et al. [13] demonstrated that the wear behavior of a honed cutting edge strongly depends on the micro-geometry of the honed edge during the slot milling of a quenched and tempered 42CrMo4 steel using cemented carbide inserts. The results showed that the tool wear type shift from a flank wear to a rake face wear depending on a defined micrometrical parameter named the “form factor K”. The authors have also shown that the tool life of sharp cutting edges is limited by chipping wear. This concept was adopted by Fulemova and Janda [14] for studying the cutting forces and cutting tool life when milling ferrite-martensite steel (EN ISO X12CrMoVNbN9-1) using sub-micron sintered carbide tools. The authors pointed out that an edge radius of 15 µm is optimal in terms of tool wear and cutting forces. Conversely, Elbestawi et al. [8] showed that sharp edge outperforms both honed and chamfered PCBN cutting tool edges during the hard milling of AISI H 13 steel.

Despite their introduction into the machining industry in the early 1970s, the use of ceramic cutting tools for hard milling remains very limited compared to the turning process [15]. The fundamental idea behind the use of the ceramic tool is to take advantage of their high-temperature resistance, reduced costs and achieve increased material removal rates during machining of hardened alloys [16]. A recent study by Wang et al. [17] investigated the cutting tools performances of four solid ceramic end milling tools as well as Ti (C, N), Si_3_N_4_, SG4 and LT55 in machining hardened AISI H13 steel (60–62 HRC). Wang investigated that the cutting forces of ceramic end milling tools are smaller than cemented carbide tool, and such ceramic tools of Si_3_N_4_, Ti (C, N) and LT55 gave the best surface finish and had a longer tool life. Koshy et al. [18] investigated the performances of two different cutting tools, including an indexable ball nose end mills (with carbide and cermet insert), and a solid carbide ball nose end mills, in face milling of hardened D2 tool steel (58 HRC). Results showed that cutting performance of an indexable cermet insert is lesser than both types of cemented carbide tools. However, several studies demonstrated the encouraging potential of ceramic tools in producing favorable surface integrity [6,19] and reasonable tool life [20,21,22,23] at relatively high cutting speed when turning hardened steel. Research studies have shown that ceramic tools are significantly superior to carbide tools [24] and in some cases can outperform CBN/PCBN tools [22] in terms of cutting tool life.

In addition to tool materials and edge preparation, the work material hardness effect during hard machining was also the subject of some research studies. However, only a few focused on the hard milling process. In investigating the influence of work material hardness in high-speed milling of hardened AISI H13 steel (45 HRC and 55 HRC), Elbestawi et al. [8] showed that the higher the workpiece hardness, the lower the wear on high CBN cutting tools. Wang and Zheng [25] studied the ball end milling of hardened AISI H13 steel (20 HRC and 41 HRC) using a TiAlN coated carbide tool. Results showed that the specific chip shearing energy and friction coefficient were greater for the soft/ductile material (20 HRC). On the other hand, several studies focusing on the effect of work material hardness during hard turning were carried out. Poulachon et al. [26] reported the existence of a limiting value of hardness at 50 HRC in hard turning 100Cr6 steel using PCBN tools. Below this limit, the cutting forces decrease, and the wear resistance is controlled by the bonding strength of the tool grains. Above the limit of 50 HRC, the cutting forces increases, and abrasion is the main tool wear mechanism. Furthermore, results showed equivalence between cutting speed and work material hardness. These results were in agreement with those obtained by Ng and Aspinwall [27] for hard turning AISI H13 steel. Results reported by Thiele and Melkote [28] gave evidence of a significant interaction between edge preparation and work material hardness during hard turning AISI 52,100 steel. Nevertheless, most of the findings in hard turning were not yet validated in hard milling.

The AISI 1045 steel is also heat-treated steel, which is well used in the manufacturing of several structural components for different industrial applications. Although it is considered as the most studied steels from the machining point of view, the AISI 1045 steel was rarely investigated under hard milling conditions using ceramic tools [1,2,3,4,15]. In fact, most of the studied AISI 1045 steel have annealed microstructures with hardness not exceeding 234 HB (22 HRC) and milled using carbide tools [3,29,30,31,32,33,34,35]. To this end, the present paper aims to explore the performance of ceramic tools in hard milling of AISI 1045 steel. In addition to standard technological parameters such as cutting speed, feed rate and milling configuration, the study also focuses on the specific effects of cutting-edge preparation and work material hardness on machinability indexes, including cutting forces, cutting power and tool wear under dry machining conditions. A future article will focus on ecological aspects, especially on fine and ultrafine particles emission.

## 2. Methodology

To study the hard milling of the AISI 1045 steel, a standard orthogonal array design of experiments (DoE) named L32 (2^1^ 4^4^) was employed. This DoE was selected due to its ability to identify the main effects of the factors and their interactions. The factors’ levels and the design matrix are given in Table 1 and Table 2, respectively. All the experiments were carried out under dry condition using fixed axial (a_p_ = 2 mm) and radial (a_e_ = 25.4 mm) depths of cut.

The proposed DoE allows studying the effects of machining technological parameters on machinability indicators: cutting force, cutting power and tool wear.

### 2.1. Resultant Cutting Force (F)

In the milling process, different cutting force components can be extracted from force signals. Ceramic inserts are significantly sensitive to vibration phenomenon, which in turn depends on the cutting forces in the *x*-direction ($F_{x}$) and *y*-direction ($F_{y}$; Figure 1). Hence, the maximum resultant cutting force F (Figure 2b) operating in the (*x, y*) plane was adopted in the present study. F is given by the following formula:
(1)$$F\left(j,t\right)=\sqrt{Fx{\left(j,t\right)}^{2}+Fy{\left(j,t\right)}^{2}}.$$

An illustration of *x* and *y* forces and their resultant force F(t) signals are depicted in Figure 2a,b, respectively. In the present study, the resultant force F used in the statistical analysis is calculated as follows:
(2)$$F=\frac{\sum_{j=1}^{4}F_{max}\left(j,t\right)}{4}$$
where $F_{max}\left(j,t\right)$ is the maximum values of the resultant force recorded for the *j*th insert.

### 2.2. Cutting Power (Pc)

The cutting power (*P_C_*) is a relevant indicator of the total power consumption during the machining. It depends on the cutting speed (*v_c_*) and the tangential cutting force component (*F_t_*). The later should be extracted from the cutting forces measured in the *x*, *y* and *z* directions (Figure 1). The relationship between the global coordinate system linked to the machine tool (*x*, *y*, *z*) and the local coordinate system linked to the insert (*t*, *r*, *a*) is defined as follows:
(3)$$\left(\begin{array}{c} F_{x}\left(j,\theta \right)\\ \begin{array}{c} F_{y}\left(j,\theta \right)\\ F_{z}\left(j,\theta \right) \end{array} \end{array}\right)=\left(\begin{array}{ccc} \cos\theta & \sin\theta & 0\\ \sin\theta & -\cos\theta & 0\\ 0 & 0 & 1 \end{array}\right)\left(\begin{array}{c} F_{t}\left(j,\theta \right)\\ \begin{array}{c} F_{r}\left(j,\theta \right)\\ F_{a}\left(j,\theta \right) \end{array} \end{array}\right)=A\left(\begin{array}{c} F_{t}\left(j,\theta \right)\\ \begin{array}{c} F_{r}\left(j,\theta \right)\\ F_{a}\left(j,\theta \right) \end{array} \end{array}\right),$$
where θ is the tool rotation angle. $F_{t}$, $F_{r}$ and $F_{a}$ are the tangential, radial, and axial cutting forces, respectively. They can be calculated by the following equation:
(4)$$\left(\begin{array}{c} F_{t}\left(j,\theta \right)\\ \begin{array}{c} F_{r}\left(j,\theta \right)\\ F_{a}\left(j,\theta \right) \end{array} \end{array}\right)=A^{-1}\left(\begin{array}{c} F_{x}\left(j,\theta \right)\\ \begin{array}{c} F_{y}\left(j,\theta \right)\\ F_{z}\left(j,\theta \right) \end{array} \end{array}\right).$$

Then, the cutting power $P_{c}$ (kW) can be calculated as follows:
(5)$$P_{c}=V_{c}\sum_{j=1}^{n}F_{tmax}\left(j\right),$$
where $F_{tmax}\left(j\right)$ is the maximum values of the tangential force recorded for the *j*th insert. *n* denotes the number of inserts operating simultaneously during the machining.

## 3. Experimental Procedure

### 3.1. Work Material

The work material was a medium carbon steel AISI 1045 widely used in the fabrication of hydraulic shaft, pump shaft, piston rods, cylinders, cams and crankshafts. The dimensions of the AISI 1045 rectangular block used were 250 mm (length) by 100 mm (width) and 25 mm (thickness). Four varying hardness were employed in the tests. The hardening process consisted of heating the part to an austenitization temperature of 844 °C for 1.5 h and followed by quenching in water and tempering in the range between 260–470 °C for 2 h, depending on the targeted hardness value (38, 43, 48 and 53 HRC). Table 3 lists the percentage weight chemical composition of the AISI 1045 steel.

### 3.2. Face Milling Tests

The experimental trials were carried out on a MAZAK NEXUS 410A vertical CNC milling machine (Yamazaki Mazak Corporation, Oguchi, Japan, Max. RPM = 12,000 rev/min, power at 5000 rpm = 25 HP) under dry machining conditions (Mazak, Oguchi, Japan). Round ceramic inserts with two distinct edge preparations (T-land and honed) and of two different grades (KY2100 and KY4300) were used in the machining tests as indicated in Table 4. A shell mills type KDNR250RN40C3 tool holder with four inserts from Kennametal, Latrobe, PA, USA) was also used. The physical and mechanical properties of the workpiece (AISI 1045 steel) and the cutting inserts are depicted in Table 5. The assembled tool (cutter plus four inserts) was balanced before the machining process.

Cutting forces measurement was acquired using the Kistler 9255-B three-component dynamometer (Kistler^®^, Winterthur, Switzerland) at a sampling rate of 48 kHz (Figure 3). These were later processed used the MATLAB^®^ software (version r2017b, MathWorks^®^, Natick, MA, USA). After machining, tool wear measurement was performed using the KEYENCE VHX-500 FE Digital Microscope (Keyence, Osaka, Japan).

## 4. Results and Discussions

### 4.1. Analysis of Resultant Force

The aim of the analysis of variance (ANOVA) was to investigate which of the cutting parameters significantly affect the above-mentioned performance characteristics. This is accomplished by separating the total variability of the mean ratios, which is measured by the sum of the squared deviations of all the observations, from their mean to the total mean, into representing it as contributions of each cutting parameters as well as the error.

The total sum of the squared (TSS or SST) deviations comprises of two sources: the sum of the squared deviations due to each cutting parameter (SSd) and its interaction effects and the sum of the squared error (SSe). The percentage contribution by each of the cutting parameter in the total sum of the squared deviations SST can be used to evaluate the importance of a change in that cutting parameter on a specific performance characteristic evaluated. In addition, the F-test named in honor of Sir Ronald Aylmer Fisher is also used to statistically test the equality of means and test the overall significance for a regression model. The probability from the F-statistics allowed us to determine if the model met the null hypothesis. A statistically significant result was found at high F-value and when a probability (*p*-value) was less than a pre-specified threshold (significance level), commonly defined as 0.05 (95%).

The analysis of variance (ANOVA) of the resultant force is shown in Table 6. It can be seen that the feed rate (F-value = 33.43) was the most significant to the output response of the resultant force, followed by the interaction milling type × tool (F-value = 3.55). The hardness, the tool, the milling type and cutting speed had no significant effect on the resultant force. The contribution (%) provided an additional depiction of significance for the interpretation of the results. The results show that the contribution due to the feed rate was 74.58%, whereas the interaction milling type × tool contributes 7.91%, the hardness contributed 3.91%, the tool contributed 2.76% and the milling type interaction speed contributed 1.36%.

An analysis of the mean response of the resultant force is shown in Table 7. The table shows the optimal levels of the control factors for the resultant force values performed by the Taguchi method. These optimal values are also presented in Figure 4. The optimal cutting parameters for minimizing the resultant force could be clearly determined from this figure. The levels and mean ratios for the factors that give the best resultant force were MT (Level 1, mean = 435.7), *v_c_* (Level 4, mean = 423.7), *f_n_* (Level 1, mean = 249.4), H (Level 1, mean = 394.5) and CT (Level 1, mean = 393.1). In other words, an optimum resultant force value was obtained during up the milling type at 500 m/min cutting speed, 0.05 mm/tooth feed rate, 38 HRC material hardness and using a honed silicon nitride (SiAlON) base ceramic tool. From Figure 4, the use of high cutting speed and low feed rate generated lower cutting forces [42,43,44,45,46,47,48,49]. The decrease in cutting force from higher cutting speed was due to the rise in temperature in the shear plane area, which lowers the shear strength of the material [44]. In addition, the increase in the feed rate induced a larger volume of the cut material in the same unit of time, besides establishing a dynamic effect on the cutting forces. It also led to a corresponding increase in the normal contact stress at the tool chip interface and in the tool chip contact area [48,50,51]. From Figure 4, the tool grade material composition further compounded the influence of the honed edge preparation. Conversely, this property bears little or no effect on T-land edge preparation. The maximum resultant force was obtained at about 43 HRC. Further increase in material hardness (48–53 HRC) led to a decrease in the resultant force. This phenomenon has also been previously reported by Matsumoto et al. [52] and Chao and Trigger [53].

The low delta value for the milling type means response further indicates the low effect milling kinematics bares in resultant force prediction. Experimental test in optimal conditions for other factors and using down milling generated similar/improved results. This is believed to be attributed to the gradual reduction in chip thickness and improved chip flow in this process.

### 4.2. Analysis of Power

The analysis of variance (ANOVA) of the cutting power is shown in Table 8. From the table, it can be seen that the feed rate (F-value = 7.45) was the most significant terms related to the cutting power, followed by the cutting speed (F-value = 6.96). The table shows that the milling type, hardness and tool were not significant terms to the cutting power. The contributions of milling type, cutting speed, feed rate, hardness, tool, milling type × speed and milling type × tool on power were found to be 0.204%, 33.78%, 36.13%, 1.72%, 1.74%, 1.18% and 5.82%, respectively. Thus, an equivalent importance of the feed rate (36.13%) and cutting speed (33.78%) were observed to cutting power performance. Similar results were reported by Benlahmidi et al. [54] when turning hardened AISI H11 steel (50 HRC) with CBN7020 tools. They reported that the cutting speed, feed rate and depth of cut are the most significant terms on cutting power. This was also confirmed by Davim and Figueira [55] during the turning AISI D2 steel using traditional and wiper cutting tools. In milling processes, Fratila et al. [56] indicated that the increase in cutting power was associated with a corresponding increase of cutting speeds, feed rates and depth of cuts, when machining AlMg_3_ with HSS (high-speed steel) tools. This was also observed by Pa Nik et al. [46] who indicated an increase in cutting power with an increase in cutting conditions (feed rate and depth of cut) in both dry and wet machining conditions.

The various levels of cutting parameters to minimize cutting power are given in Table 9 and their main effect plot is shown in Figure 5. The optimal parametric combination for minimized power was at MT (Level 1, mean = 2154), *v_c_* (Level 1, mean = 1310), *f_n_* (Level 1, mean = 1310), H (Level 1, mean = 2038) and CT (Level 4, mean = 2032). Thus, an optimum power value was obtained during the up milling type, at 200 m/min cutting speed, 0.05 mm/tooth feed rate, 38 HRC hardness and, using a chamfered tool with a matrix of Al_2_O_3_ + SiCW.

The low delta value between MT mean responses highlights the reduced influence of milling kinematics in power estimation. Further experimental tests at optimal settings and using down milling also resulted in lower power values. The gradual reduction in chip thickness and the improved chip evacuation from the tool cutting zone was believed to be responsible for these results.

### 4.3. Analysis of Tool Wear

Tool wear investigations showed that flank wear and micro-chipping occurred due to abrasion of hard particles and the effect of recurring tool entry into the workpiece (Figure 6). No build-up edge (BUE) was observed during the experiments. This can be explained by the fact that the cutting speed used with ceramic is usually high. From the literature, the increase in alumina content improves the dissolution resistance of the material but this is prone to local plastic deformation at high temperatures [57,58]. Figure 7 shows a preview into the SEM microstructure of a worn part of the KY2100 grade insert with a composition of silicon nitride (SiAlON) and KY4300 with Al_2_O_3_ matrix reinforced with SiC whiskers. From the figure, the micro-shearing process and dissolution of the tool in KY2100 can be observed. Alternatively, the microstructural texture on a worn area of KY4300 tool grade shows a more compact topography, which is often more influenced by micro-scale abrasion. This is believed to be primarily as a result of the Al_2_O_3_ composition in the ceramic material, which improves its dissolution resistance during cutting.

The analysis of variance (ANOVA) of the tool wear is shown in Table 10. The analysis shows that the milling type (F-value = 6.48), tool (F-value = 6.36) and their interaction (milling type × tool, F-value = 4.12) were the significant terms to the tool wear. The hardness, the feed rate, the cutting speed and the other interactions had no significant effect on the tool wear. The most important factors affecting the tool wear were from the cutting tool, the milling type and their interactions with percentage contribution values of 31.54%, 20.42% and 10.71% respectively.

An analysis of the mean response for the tool wear, which is made by using the Taguchi method, is shown in Table 11 and Figure 8, which further highlights the significance of the cutting tool and milling type to wear. These results were in agreement with research carried out by Bouzakis et al. [59], which has shown that the kinematics of the milling process (up-milling or down-milling), considerably affects the achievable cutting performance. Hadi et al. [60] also demonstrated an increasing relationship between tool wear and up-milling kinematics. However, owing to the brittle nature of ceramics, there exists a combined effect of milling type and tool edge geometry for the wear generation in ceramics (Table 10). Severe chipping wear was often seen on honed tools with low hardness. This extreme wear is believed to be as a result of the sensitivity of the tool edge strength during impact with the workpiece, as a radial edge formation is more susceptible to impact fragmentation. This condition was reduced in tools with reinforced SiC whiskers. It is postulated that the high impact force on the lower end of the honed curve on the rake face is responsible for flacking. Optimal tool wear value was obtained with an up milling type at 500 m/min cutting speed, 0.17 mm/tooth feed rate, 48 HRC hardness and using a honed ceramic tool with reinforced SiC whisker.

## 5. Confirmation Tests

Following the determination of the individual optimal levels for each performance characteristics, a prediction and comparison of the percentage improvement of each was performed [61]. The optimal conditions of each performance characteristics (resultant cutting force (F), power (P) and tool wear (Vb)) are shown in Equations (6)–(8) below:
F_opt_ = γ_F_ + (MT_1_ − γ_F_) + (*v_c_*_4_ − γ_F_) + (*f_n_*_1_ − γ_F_) + (H_1_ − γ_F_) + (CT_1_ − γ_F_),(6)
P_opt_ = γ_P_ + (MT_1_ − γ_P_) + (*v_c_*_1_ − γ_P_) + (*f_n_*_1_ − γ_P_) + (H_1_ − γ_P_) + (CT_4_ − γ_P_),(7)
Vb_opt_ = γ_Vb_ + (MT_1_ − γ_Vb_) + (*v_c_*_4_ − γ_Vb_) + (*f_n_*_4_ − γ_Vb_) + (H_3_ − γ_Vb_) + (CT_3_ − γ_Vb_),(8)
where (F_opt_, P_opt_ and Vb_opt_) represent the optimum level average values of (F, P and Vb) from Table 7, Table 9 and Table 11, respectivley. The γ_F_, γ_P_ and γ_Vb_ state the average of all F, P and Vb values obtained from the experimental study (Table 7, Table 9 and Table 11), which are 442 N, 2213 Watt and 47.66 µm, respectively. From the calculations, it was estimated that F_opt_ = 128 N, P_opt_ = −44 Watt and Vb_opt_ = 6.8 µm. Figure 9, Figure 10 and Figure 11 show the comparison of the predicted cutting forces, power and tool wear as a function of workpiece hardness and milling modes.

In Figure 11, the decrease in predicted wear with the increase in workpiece hardness was believed to be a result of the poor chip breakability during cutting operations. Cutting chips tend to remain longer at the tooltip area in materials with lower hardness and this causes an increase in temperature and workpiece hardening. This further promotes the wear formation observed. On the other hand, good chip evacuation was observed in harder materials with a marginally lower wear occurrence. At low hardness levels (38–43 HB) the expected effect of hardness on tool wear was observed (higher hardness leading to high wear level). The same tendency is noticed for workpieces with high hardness values (48–53 HB), which generated short chips. Increasing the hardness from 48 to 53 HB led to higher cutting tool wear on both honed and chamfered inserts.

A representation of the recommended parametric combination and the initial process combination is shown in Table 12.

Figure 12 indicates an improvement from the initial process parameters when using the newly identified experimental values. An average improvement percentage above 50% is obtained in each of the evaluated responses. However, more ample work is required to establish predictive models for each of these responses. Future research will seek to develop these predictive models and estimate optimal parameters in the milling of AISI 1045 stainless steel using ceramic cutting tools.

## 6. Conclusions

This study explored the performance of ceramic tools in hard milling of AISI 1045 steel. It focused on parameters such as cutting speed, feed rate and milling configuration, as well as the specific effects of cutting-edge preparation and work material hardness on cutting forces, cutting power and tool wear. For this investigation, a Taguchi experimental design was used. From the results obtained the following conclusions were drawn:
The feed rate was the most influential parameter related to the resultant force. A decrease in feed yielded significant lower resultant forces during cutting. This parameter had a 74.58% percentage contribution to this response factor.Lower power demands were mostly influenced by the feed rate and cutting speed. These were the most important factors affecting the cutting power with comparable percentage contributions of 36.13% and 33.78%, respectively. A reduction in both cutting speed and feed rates significantly lowered cutting power.Conversely to other response factors, the choice of the cutting tool and the milling kinematics were much more important in the generation of tool wear. There exist a complex inter-correlation between the choice of cutting tool and the milling kinematics. Despite the increased impact from tool entry during down milling, response factors such as resultant force, and power could be mitigated primarily from cutting parameters (cutting speed and feed rate). Therefore under these controlled parametric conditions, the tool wear response is mostly due to tool properties (material and edge preparation). The combined influence of the choice of tool material and edge preparation had the greatest percentage contribution to flank wear formed with 31.54%. This is attributed to the sensitivity of the tool edge strength during impact with the workpiece. Honed tools with a radial edge formation were more susceptible to impact fragmentation but this could be mitigated by the strengthened presence of SiC whiskers in the KY4300 tool grade Consequently, the honed Tool T3 (RNG45E) KY4300 performed best in experimentation due to the reduced micro-chipping observed from its whisker ceramic reinforced tool edge.The material hardness in the range of 38–53 HRC was identified as not been a determinant factor on the machinability index studied in this research. Various contrasting observations of the influence of hardness were identified in the study. However, the adverse effects peaked at a hardness value of 43HRC across responses. Improved results were often identified at hardness extremes in the range studied. This can be explained by the reduction in abrasion wear between the tool and the workpiece at reduced hardness and the fall in micro-chipping wear at elevated hardness.The experimental analysis of the process yielded improved performance. The confirmation test results showed an average improved percentage above 50% for each of the response factors i.e., resultant force, cutting power and tool wear.

## Figures and Tables

**Figure 1 materials-12-03974-f001:**
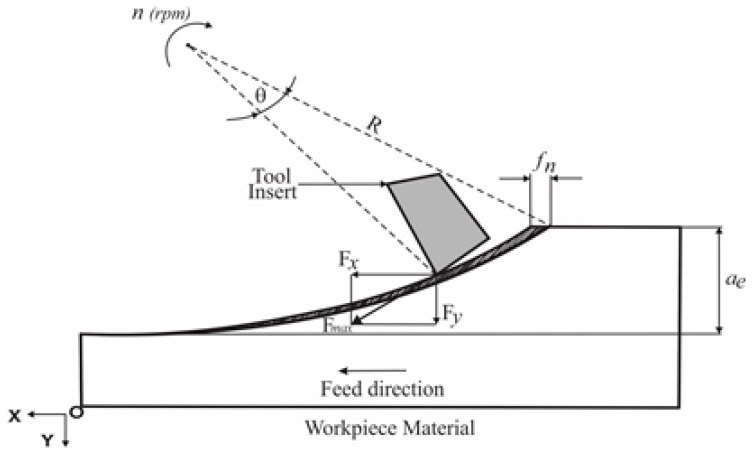
Illustration of milling force components in the cutting zone.

**Figure 2 materials-12-03974-f002:**
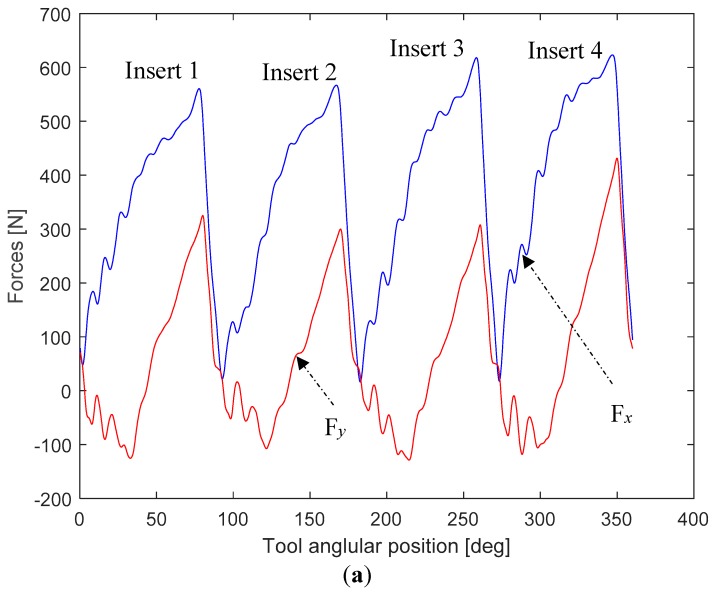
(**a**) Milling forces’ signals in the *x* and *y* directions, and (**b**) resultant force signal for machining test 8.

**Figure 3 materials-12-03974-f003:**
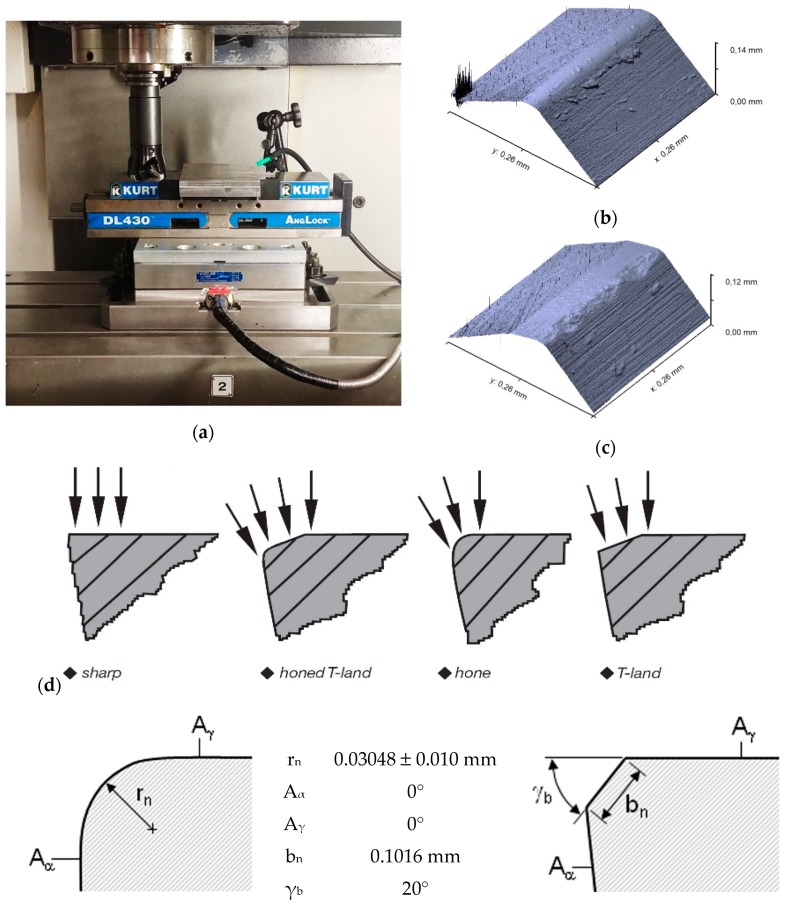
(**a**) Face milling setup with the cutting tool and force dynamometer, and micro-geometry of the (**b**) honed and (**c**) chamfered (T-land) cutting edge and (**d**) cutting edge forms with parametric edge dimensions (adapted from [36]).

**Figure 4 materials-12-03974-f004:**
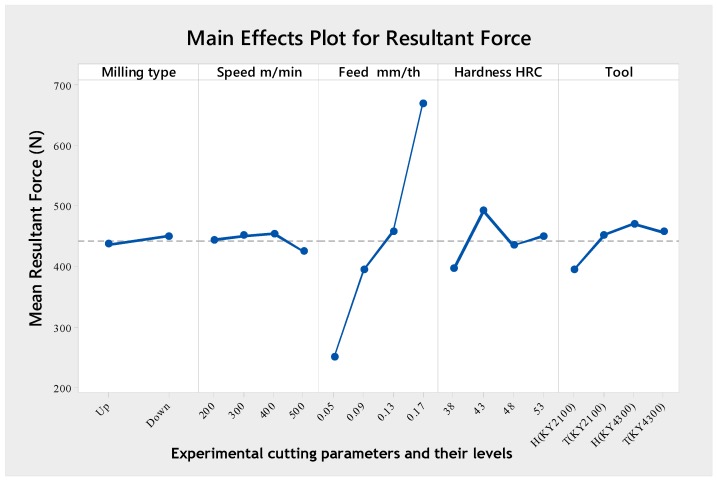
Main effects plot for resultant force.

**Figure 5 materials-12-03974-f005:**
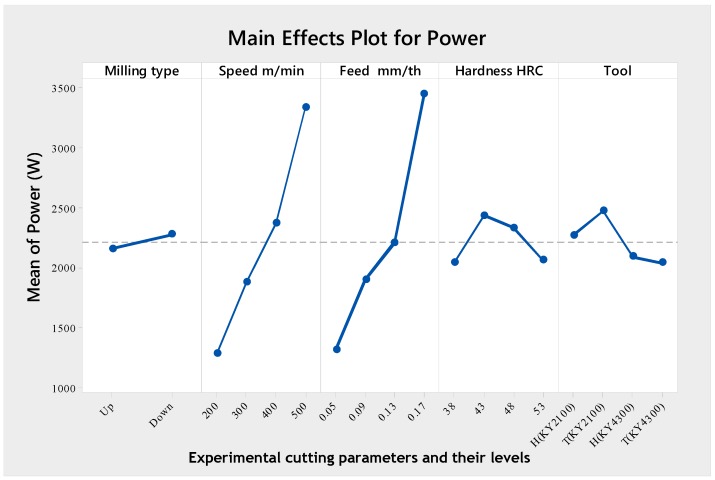
Main effects plot for power.

**Figure 6 materials-12-03974-f006:**
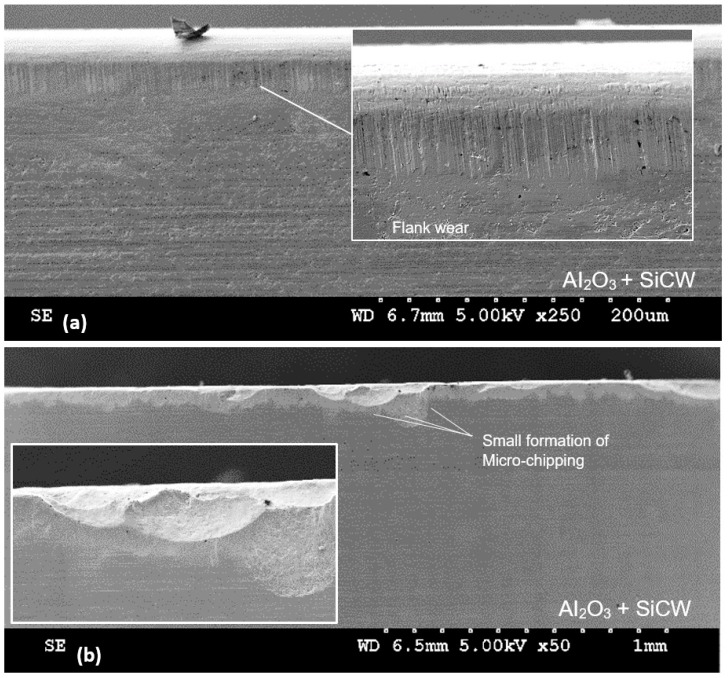
Sample wear observed on honed and chamfered tool edge preparation at increased hardness (53 HRC). (**a**) Test 8 and (**b**) test 4.

**Figure 7 materials-12-03974-f007:**
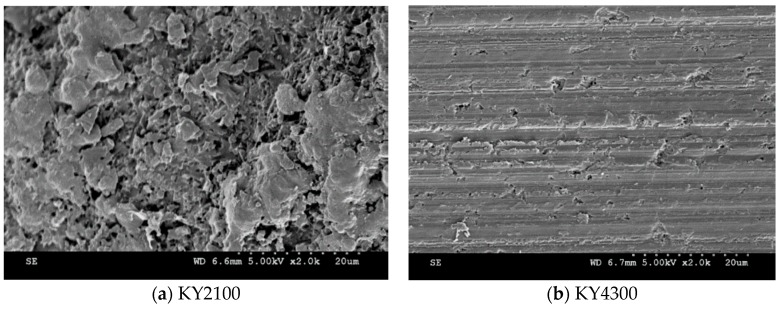
SEM image of the microstructural morphology of used ceramic inserts.

**Figure 8 materials-12-03974-f008:**
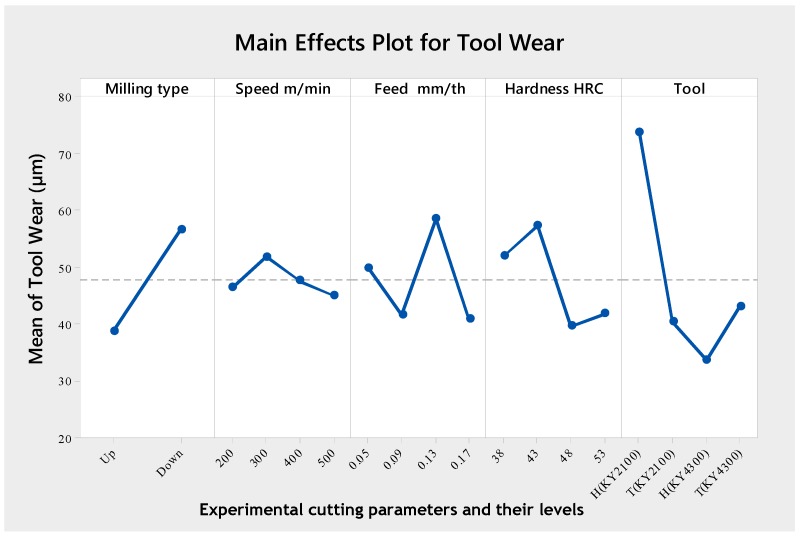
Main effects plot for tool wear.

**Figure 9 materials-12-03974-f009:**
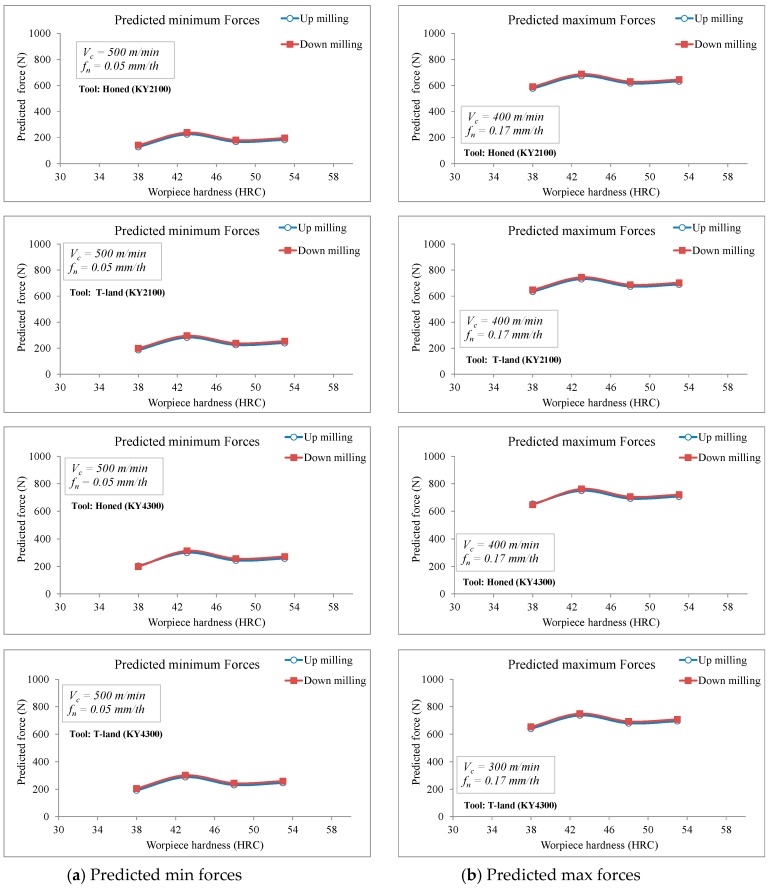
Comparison of the predicted cutting forces as a function of workpiece hardness and milling modes.

**Figure 10 materials-12-03974-f010:**
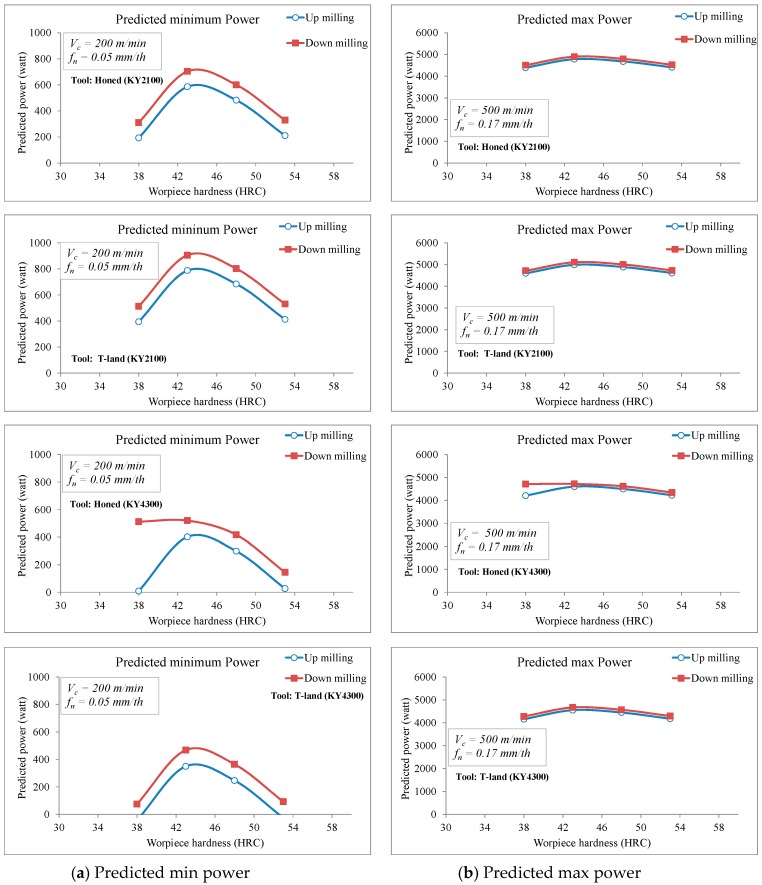
Comparison of the predicted power requirements as a function of workpiece hardness and milling modes.

**Figure 11 materials-12-03974-f011:**
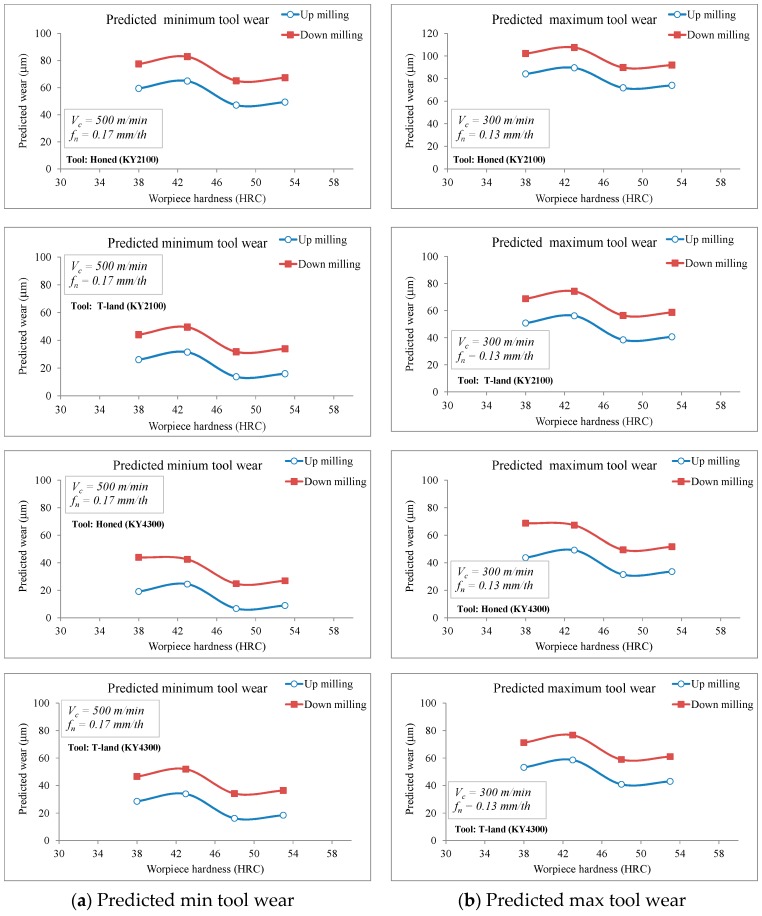
Comparison of the predicted tool wear as a function of workpiece hardness and milling modes.

**Figure 12 materials-12-03974-f012:**
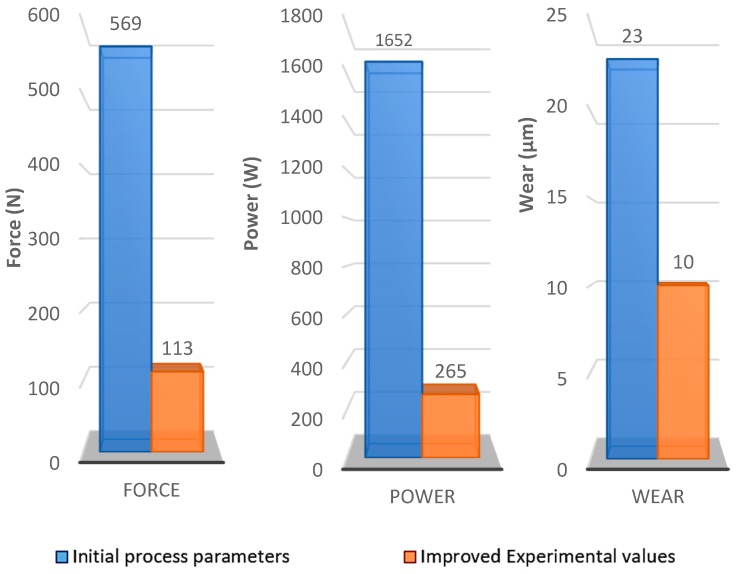
Performance chart between initial process parameters and improved experimental values for each response factors.

**Table 1 materials-12-03974-t001:** Matrix of experiments.

*Cutting Parameters*	*Level 1*	*Level 2*	*Level 3*	*Level 4*
*MT: Milling Type*	Up	Down	---	---
*v_c_: Cutting Speed* (*m/min*)	200	300	400	500
*f_n_: Feed rate* (*mm/th*)	0.05	0.09	0.13	0.17
*H: Hardness* (*HRC*)	38	43	48	53
*CT: Cutting Tools*	T1	T2	T3	T4

**Table 2 materials-12-03974-t002:** Orthogonal array L32 (2^1^ 4^4^) of Taguchi design.

No.	Milling Type	Cutting Speed (m/min)	Feed Rate (mm/th)	Hardness (HRC)	Tool
1	Up	200	0.05	38	T1
2	Up	200	0.09	43	T2
3	Up	200	0.13	48	T3
4	Up	200	0.17	53	T4
5	Up	300	0.05	38	T2
6	Up	300	0.09	43	T1
7	Up	300	0.13	48	T4
8	Up	300	0.17	53	T3
9	Up	400	0.05	43	T3
10	Up	400	0.09	38	T4
11	Up	400	0.13	53	T1
12	Up	400	0.17	48	T2
13	Up	500	0.05	43	T4
14	Up	500	0.09	38	T3
15	Up	500	0.13	53	T2
16	Up	500	0.17	48	T1
17	Down	200	0.05	53	T1
18	Down	200	0.09	48	T2
19	Down	200	0.13	43	T3
20	Down	200	0.17	38	T4
21	Down	300	0.05	53	T2
22	Down	300	0.09	48	T1
23	Down	300	0.13	43	T4
24	Down	300	0.17	38	T3
25	Down	400	0.05	48	T3
26	Down	400	0.09	53	T4
27	Down	400	0.13	38	T1
28	Down	400	0.17	43	T2
29	Down	500	0.05	48	T4
30	Down	500	0.09	53	T3
31	Down	500	0.13	38	T2
32	Down	500	0.17	43	T1

**Table 3 materials-12-03974-t003:** The chemical composition of workpiece material (% weight).

C	Mn	P	S	Si	Fe
0.459	0.721	0.0086	0.0027	0.259	balance

**Table 4 materials-12-03974-t004:** Specification of the cutting tool.

Tool ID	Edge Preparation	Kennametal Grade Name	Grade Description
1	Honed (RNG45E)	KY2100	Silicon Nitride (SiAlON) base Ceramic
2	T-land (RNG45T0420)		
3	Honed (RNG45E)	KY4300	Whisker ceramic with a matrix of Al_2_O_3_ + SiCW
4	T-land (RNG45T0420)		

**Table 5 materials-12-03974-t005:** Workpiece and cutting tool material properties [37,38,39,40,41].

Material	Density *ρ* (kg/m^3^)	Thermal Conductivity *K* (W/m-°C)	Specific Heat *C* (J/kg-°C)	Fracture Toughness *K_Ic_* (MPa-m^1/2^)	Hardness (HV)
AISI 1045	7844	45	3.7	-	-
Silicon Nitride (SiAlON) base Ceramic	3320	11.5	2.41	6.8	1650
Whisker ceramic with a matrix of Al_2_O_3_ + SiCW	3750	24	3.6	8	2000

**Table 6 materials-12-03974-t006:** ANOVA for resultant force.

SOURCE	DF	SEQ SS	ADJ SS	ADJ MS	F	P	C (%)
MT: MILLING TYPE	1	1325	1325	1325	0.18	0.677	0.135
*v_c_*: CUTTING SPEED (m/min)	3	4100	4100	1367	0.19	0.903	0.418
***f_n_*: FEED RATE (mm/th)**	**3**	**730,554**	**730,554**	**243,518**	**33.43**	**0.00**	**74.58**
H: HARDNESS (HRC)	3	38,305	38,305	12,768	1.75	0.210	3.91
CT: CUTTING TOOL	3	26,990	26,990	8997	1.23	0.340	2.76
MT × *v_c_*	3	13,376	13,376	4459	0.61	0.620	1.36
**MT × CT**	**3**	**77,516**	**77,516**	**25,839**	**3.55**	**0.048**	**7.91**
RESIDUAL ERROR	12	87,425	87,425	7285			8.92
TOTAL	31	979,591					100

Bold values show the most significant factors

**Table 7 materials-12-03974-t007:** Means response for resultant force.

LEVEL	MT	*v_c_* (m/min)	*f**_n_* (mm/th)	H (HRC)	CT
**1**	**435.7**	442.2	**249.4**	**394.5**	**393.1**
**2**	448.6	449.6	392.7	491.1	450.5
**3**		453.0	457.3	434.0	468.8
**4**		**423.7**	669.1	448.9	456.1
**DELTA**	12.9	29.3	419.7	96.7	75.6

Bold values show the optimal levels of control factors.

**Table 8 materials-12-03974-t008:** ANOVA for power.

SOURCE	DF	SEQ SS	ADJ SS	ADJ MS	F	P	C (%)
MT: MILLING TYPE	1	110,499	110499	110499	0.13	0.728	0.204
***v_c_*: CUTTING SPEED (m/min)**	**3**	**18,265,243**	**18,265,243**	**6,088,414**	**6.96**	**0.006**	**33.78**
***f_n_*: FEED RATE (mm/th)**	**3**	**19,538,513**	**19,538,513**	**6,512,838**	**7.45**	**0.004**	**36.13**
H: HARDNESS (HRC)	3	932,281	932,281	310,760	0.36	0.786	1.72
CT: TOOL	3	945,689	945,689	315,230	0.36	0.783	1.74
MT × *v_c_*	3	640,173	640,173	213,391	0.24	0.864	1.18
MT × CT	3	3,148,771	3,148,771	1,049,590	1.20	0.351	5.82
RESIDUAL ERROR	12	10,495,278	10,495,278	874,606			19.41
TOTAL	31	54,076,445					100

Bold values show the significant factors.

**Table 9 materials-12-03974-t009:** Means response for power.

LEVEL	MT	*v_c_* (m/min)	*f_n_* (mm/th)	H (HRC)	CT
1	**2154**	**1275**	**1310**	**2038**	2268
2	2272	1873	1890	2431	2469
3		2366	2206	2328	2084
4		3338	3447	2056	**2032**
DELTA	118	2062	2137	394	438

Bold values show the optimal levels of control factors.

**Table 10 materials-12-03974-t010:** ANOVA for tool wear.

SOURCE	DF	SEQ SS	ADJ SS	ADJ MS	F	P	C (%)
**MT: MILLING TYPE**	**1**	**2601.2**	**2601.2**	**2601.19**	**6.48**	**0.026**	**10.71**
*v_c_*: CUTTING SPEED (m/min)	3	214.4	214.4	71.48	0.18	0.909	0.88
*f_n_*: FEED RATE (mm/th)	3	1667.9	1667.9	555.96	1.39	0.295	6.86
H: HARDNESS (HRC)	3	1695.9	1695.9	565.30	1.41	0.288	6.98
**CT: TOOL**	**3**	**7658.9**	**7658.9**	**2552.96**	**6.36**	**0.008**	**31.54**
MT×SPEED	3	672.9	672.9	224.30	0.56	0.652	2.77
**MT×TOOL**	**3**	**4959.0**	**4959.0**	**1653.02**	**4.12**	**0.032**	**20.42**
RESIDUAL ERROR	12	4815.1	4815.1	401.26			19.83
TOTAL	31	24,285.3					100

Bold values show the most significant factors.

**Table 11 materials-12-03974-t011:** Means response for tool wear.

LEVEL	MT	*v**_c_* (m/min)	*f_n_* (mm/th)	H (HRC)	CT
**1**	**38.64**	46.29	49.87	51.93	73.78
**2**	56.67	51.84	41.52	57.35	40.41
**3**		47.58	58.52	**39.58**	**33.52**
**4**		**44.93**	**40.72**	41.78	42.92
**DELTA**	18.03	6.91	17.79	17.77	**40.26**

Bold values show the optimal levels of control factors.

**Table 12 materials-12-03974-t012:** Initial and recommended parameters combination.

Performance Characteristics	Initial Process Parameters Levels	Recommended Process Parameters Levels
Resultant force (F)	MT_1_ *v_c_*_1_*f_n_*_4_H_4_CT_4_	MT_2_ *v_c_*_4_*f_n_*_1_H_1_CT_1_
Power (P)	MT_1_ *v_c_*_1_*f_n_*_4_H_4_CT_4_	MT_2_ *v_c_*_1_f*_n_*_1_H_1_CT_4_
Tool wear (Vb)	MT_1_ *v_c_*_1_*f_n_*_4_H_4_CT_4_	MT_1_ *v_c4_f_n_*_4_H_3_CT_3_

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
