# Peer review of "Machinability Study of Hardened 1045 Steel When Milling with Ceramic Cutting Inserts"

_materials, 2019, doi:10.3390/ma12233974_

Round 1
Reviewer 1 Report
In the study, ceramic tools were used to cut 1045 stainless steel of different hardness for detailed research and analysis. It is of great help to the industry engaged in related cutting. There are several questions for this article that need to be confirmed.
1. In line270 milling type*tool contributes “*” stands for multiplication sign? Please define clearly. The MT*VC in Table 6 also has the same problem as MT*CT. Where “VC” should be “Vc” as defined in Table 1. Please check all the symbols in the article again.
2. In Table 14, the purpose of parameter optimization should improve cutting performance. Can you explain why power in optimizes experimental data higher than the initial setting.
3. The temperature prediction data shown when Vc=500m/min, fn=0.05mm/th, and the hardness of the work piece is greater than HRC42, the predicted cutting temperature seems to be higher than the A1 temperature of 1045 stainless steel. If the material is cut under this parameter, the metallographic structure and hardness of the material surface may change.
4. Analysis of tool wear where milling type and tool type are the main factors affecting tool wear. However, it is shown from the estimated data in Fig. 13 that the tool wear has a tendency to decrease with increasing hardness. It is very puzzling that the cutting of softer material tools has a higher wear ratio. Can you have a detailed explanation?
5. The difference between the pridicted data and the experimental results of the numerical analysis is too large, as shown in Table 14. Correction or optimization should be made for the theoretical prediction analysis model to ensure the accuracy of the pridiction.
Author Response
The authors would like to tanks the reviewers for their time and for the comments that helped improving the manuscript. They will find below detailed responses to their comments.
Reviewer 1
In the study, ceramic tools were used to cut 1045 stainless steel of different hardness for detailed research and analysis. It is of great help to the industry engaged in related cutting. There are several questions for this article that need to be confirmed.
In line270 milling type*tool contributes “*” stands for multiplication sign? Please define clearly. The MT*VC in Table 6 also has the same problem as MT*CT. Where “VC” should be “Vc” as defined in Table 1. Please check all the symbols in the article again.
Symbols in the manuscript have been checked. Issues have been addressed as requested.
In Table 14, the purpose of parameter optimization should improve cutting performance. Can you explain why power in optimize experimental data higher than the initial setting.
A contradictory influence of the milling kinematics was experienced in the study due to the added influence of chip evacuation. Chips produced returned to the machined surface and negatively affected both forces and power readings. Due to the negligible effect of milling kinematics, experimental optimal parameters selected using down milling produced improved results with a better chip evacuation.
The temperature prediction data shown when Vc=500m/min, fn=0.05mm/th, and the hardness of the workpiece is greater than HRC42, the predicted cutting temperature seems to be higher than the A1 temperature of 1045 stainless steel. If the material is cut under this parameter, the metallographic structure and hardness of the material surface may change.
The analysis of variance previously done on temperature data and presented did not show any significant effect of tested factors on the response. The analysis of temperature was withdrawn for further work. This would give the authors time to identify significant factors to temperature and validate its results to confirm if actually the temperature exceeds the A1 of 1045 stainless steel.
Analysis of tool wear where milling type and tool type are the main factors affecting tool wear. However, it is shown from the estimated data in Fig. 13 that the tool wear has a tendency to decrease with increasing hardness. It is very puzzling that the cutting of softer material tools has a higher wear ratio. Can you have a detailed explanation?
This behavior can be explained by the poor effect of workpiece hardnesses and microstructures tested on tool wear. In fact, the effect of workpiece hardness on tool wear was very low, almost negligible (See Anova Table 10). Only the effects of milling types, tool and interactions with these last two factors were significant.
In the main effects plot (figure 9) this behavior is also observed. For low hardness values (38-43 HB) and high hardness (48-53), increasing the hardness from 38 to 43 HD led to increased wear; idem for hardness from 48-53. In the Figure referred by the reviewer, the authors were trying to depict the effects of cutting tools, milling types on tool wear for each of the workpiece material tested and for selected speed and feed rates. The decrease in predicted wear with increase in workpiece hardness could also be due of the poor chip breakability during cutting operations. Cutting chips tend to remain longer at the tooltip area in materials with lower hardness and this result in a temperature rise and workpiece hardening. With harder material, good chip evacuation was observed. At these low hardnesses (38-43HB) the expected effect of hardness on tool wear was observed (higher hardness leading to high wear level).
A text relative to this effect of chip was added to the manuscript.
The difference between the predicted data and the experimental results of the numerical analysis is too large, as shown in Table 14. Correction or optimization should be made for the theoretical prediction analysis model to ensure the accuracy of the prediction.
The interference experience from chip evacuation bears responsibility to some notable differences. The authors identify that, with alternate milling type and identified good values for other factors, improved values are obtained. They are trying to show how by using the data obtained, low cutting forces, power and tool wear could be obtained.
The manuscript was modified to put emphasis on the difference between original data and those obtained after experimental confirmation tests following analysis.
As the paper predominantly focusses on the machinability, the authors have indicated that future research areas will focus on establishing models related to the predictions of these response factors.
The authors

Reviewer 2 Report
Please see the attachment.

Author Response
MDPI- 1045 – Detailed responses to reviewers’ comments
The authors would like to tanks the reviewers for their time and for the comments that helped improving the manuscript. They will find below detailed responses to their comments.
Reviewer 2
Row 175: equation (1), letters x and y are subscripts
These have been modified as requested.
Row 162: figure 1 - please, check feed direction section 2.2 please, unify pc and pc cutting speed is denoted vc. Mostly, letter v is lowercase.
These have been modified as requested in all the manuscript
Row 265: ...feed rate (f=33.43) is the ... F or f? F is for force.
This has been modified as requested. “F” in this context stands for the fisher’s value and has been replaced with “F-value”
Row 424: tool wear (vb) - table 14: tool wear (vb). Please unify. Tables 6-13 would be more clear with a better explanation of abbreviations and the addition of units.
These symbols have been unified in all the manuscript. Tables 6-13 have also been modified as requested.
Question: What is the radius of curvature of the honed cutting edge in Fig.3? Maybe, it would be more appropriate to show other dimensions as in Fig. 3 (b) and (c). “more appropriate to show other dimensions” in Fig. 3b, 3c, 3d, I meant the dimensions according to the next figure. The provided dimensions of cutting edge morphology in Fig. 3b and 3c are useless.
Abdulkadir, L. N., Abou-El-Hossein, K., Jumare, A. I., Odedeyi, P. B., Liman, M. M., & Olaniyan, T. A. (2018). Ultra-precision diamond turning of optical silicon—a review. The International Journal of Advanced Manufacturing Technology, 96(1-4), 173-208.
We recognize rn and rε in cutting tool geometry in two different planes. The reader of the manuscript does not know what is rε that influencing significantly components of machining force and quality of machined surface. (The figure is for turning tool, however, the principle is similar.)
VASILKO, K. (2007). Analytická teória trieskového obrábania (Analytical theory of chip machining). Prešov: COFIN.
Vasilko, K., MurÄŤinková, Z., & NosáÄľ, J. (2019). Materials and machining trends in terms of the existing axioms of the machining theory. Materialwissenschaft und Werkstofftechnik, 50(2), 165-173.
Additional information on the tool edge preparation have been included in fig. 3. This information provides the parameters rn , Aa, Ag, bn,, gb which permit to establish the actual geometry of the edge. This also included figure to other the position on these parameters.
Rows 79-80: The authors pointed out that an edge radius of 15 μm is optimal in terms of tool wear and cutting forces.
Can we state a similar conclusion for milling with ceramic cutting inserts? We do not know dimensions of cutting wedge in presented study.
Unfortunately, the authors may not, at this stage make such a conclusion as no modification were performed on the tool edge geometry. Variations in tool geometry could not cover in this current study but is an area of interest for future work. The edge preparations varied only between honed and chamfered and were selected from commercially available vendors.
Suggestion: The manuscript provides too much data and information resulting in difficult reading. I suggest splitting the manuscript into two or more manuscripts.
The authors removed the analysis of the temperature as suggested by the reviewer. The analysis of variance previously done on temperature data did not shown any significant effect of tested factors on the response. The authors believe this would permit ton conduct additional analysis to identify significance between this response and tested factors. The work now uses three factors, resultant force, power, and wear which provide an adequate view of the machinability.
